# OpenReview forum: "NetArena: Dynamic Benchmarks for AI Agents in Network Automation"
_ICLR.cc/2026/Conference — ICLR 2026 Poster_

### Official Review · Reviewer_d7Bd · 2025-10-27

**Soundness:** 3
**Presentation:** 3
**Contribution:** 3
**Rating:** 6
**Confidence:** 1

**Summary:**

The authors propose a dynamic benchmarking framework named NETARENA, designed to evaluate the performance of LLMs in network system applications. Unlike traditional static benchmarks, NETARENA can dynamically generate unlimited queries and integrates with high-fidelity network emulators to assess correctness, safety, and latency.

**Strengths:**

1. The paper introduces a dynamic LLM benchmark generation framework specifically for the networking domain, demonstrating clear innovation.
2. Beyond traditional correctness metrics, the benchmark incorporates safety and latency as key evaluation dimensions, which better align with the needs of high-stakes systems.
3. The paper is well written and clearly presented.

**Weaknesses:**

1. Although the paper defines safety and latency evaluation standards, it lacks explicit quantitative formulas or threshold specifications.
2. The evaluation focuses on three types of network tasks, but broader validation across more diverse scenarios is missing. The authors could further discuss potential directions for future evaluation (additional experiments are not necessary).
3. While correctness, safety, and latency often involve trade-offs, the paper does not provide corresponding quantitative analyses to characterize these relationships.

**Questions:**

1. Can NETARENA support cross-task generalization testing? For example, can a model trained on routing tasks generalize to microservice policy troubleshooting tasks?
2. In practice, how much time and computational resources are required to complete a full-scale evaluation?
3. Were the dynamically generated natural language task templates reviewed or validated by human experts? How do you ensure consistency between task descriptions and the simulated system states?

---

> ### Author Response · Authors · 2025-11-21
>
> Dear Reviewer,
>
> Thank you for your constructive feedback. Below, we address your comments point by point.
>
> 1. **Explicit quantitative formulas for Safety and Latency**
>
> **Safety.** As defined in Eq. 5, if any step in an LLM’s output violates a safety policy, the output is unsafe. Thus each query yields a binary Pass/Fail outcome. Across N queries, we compute the safety score using the standard error of a Bernoulli mean, then report its 95% confidence intervals.
>
> **Latency.** For each query, we log the end-to-end inference time or iterations, and report the mean latency per application.
>
> ---
>
> 2. **Potential directions for future evaluation**
> To have a broader validation across more diverse scenarios, we are working on expanding NetArena. Below table is the summary of the future evaluation plan.
> | Application              | Current error types | More diverse errors and topology                     | More diverse metrics                                                |
> |--------------------------|------------------------------------|------------------------------------------------------------------------|--------------------------------------------------------------------------------|
> | Routing (in NetArena)    | 15 (Table 5)        | Apply error at different levels (e.g., multi-hop path error)           | Expand safety metric to evaluate cascading routing faults                     |
> | K8s (in NetArena)        | 15 (Table 6)        | Add multi-cluster topology setup                                       | Expand safety metric to detect cascading failures (e.g., service downtime)   |
> | Config generation (new)  | –                                  | Inject diverse errors (e.g., missing fields, incompatible parameters)  | Correctness: check if output config matches intent; Safety: prevent outages   |
>
> We are expanding NetArena to more models, agents and applications with evaluators. Please see the anonymized leaderboard: https://alphabetsoup628.github.io/netarena_leaderboard/
>
> For applications lacking established emulators, recent work [1] shows that LLMs combined with formal methods can automatically synthesize emulators. Integrating this idea is a natural next step to expand NetArena to a broader set of real-world environments.
>
> **[1]** Archit Bhatnagar, et al. *A Case for Learned Cloud Emulators*. In Proceedings of the 24th ACM Workshop on Hot Topics in Networks (HotNets '2025).
>
> ---
>
> 3. **Tradeoff among correctness, safety, latency**
> We run a small study on App-Routing (10 queries) where we explicitly ask Qwen-72B (CoT) agent to optimize only one metric: correctness, safety, or latency.
>
> Below table suggests that in this preliminary study, there are clear tradeoffs among metrics, while focus on safety in this case appears to be more favorable in most settings.
>
> | Objective           | Correctness | Safety | Latency|
> |---------------------|-------------|--------|-----------------------|
> | General (original)            | 14%         | 9%     | 10                    |
> | Correctness-focused | 18%         | 3%     | 15                    |
> | Safety-focused      | 18%         | 15%    | 18                    |
> | Latency-focused     | 5%          | 9%     | 4                     |
> ---
> 4. **Can NetArena support cross-task generalization testing?**
> This is an interesting question! Yes. Any fine-tuned model can be evaluated on other tasks using the same interface. However, cross-task performance is currently weak. For example, a model fine-tuned on capacity planning tends to inject irrelevant capacity-planning logic when tested on routing. We view cross-task generalization as a potential future work, and NetArena provides the infrastructure to study it systematically.
>
> ---
>
> 5. **Time and computational resources used for full-scale evaluation?**
> Below we provide concrete numbers for two agents.
> | Application          | Capacity Planning (5000 query) | Routing (2250 query) | K8s (2000 query) |
> |----------------------|-------------------------------|----------------------|------------------|
> | Input + Output Token usage on GPT-4o | 8.2 million                | 1.13 million          | 6.2 million      |
> | Computational time (Qwen-72B on 1 A100 GPU) | 18.5 hours                | 19.4 hours           | 31.0 hours       |
> ---
>
> 6. **Consistency between task descriptions and the simulated system states**
> The natural language template is indeed reviewed by human experts to ensure consistency.
> For reactive applications, the template is filled with the error messages (in natural language) from the emulator after injecting the faults, so that the language description is fully aligned with the true system state.
> For constructive applications, human experts create the natural language template for each action. During the query generation, the synthesized actions are sequentially added to the task description based on the template, which makes it semantically identical to the underlying state transitions.

---

> > ### Comment · Reviewer_d7Bd · 2025-11-24
> >
> > Thank you for the authors’ response. I will maintain my positive score.

---

> > > ### Author Response · Authors · 2025-11-26
> > >
> > > Dear Reviewer d7Bd,
> > >
> > > Thank you for your response! Just for your information, we added more details for the RL-based fine-tuning use case to the paper draft, including the new Figure 6. All the changes are marked in blue for your convenience.
> > >
> > > Please feel free to let us know if you have any more questions or concerns! We would love to address them further.
> > >
> > > Best wishes to your work!
> > >
> > > -NetArena Authors

---

### Official Review · Reviewer_kJeZ · 2025-11-01

**Soundness:** 3
**Presentation:** 3
**Contribution:** 3
**Rating:** 6
**Confidence:** 3

**Summary:**

The paper introduces a framework, NETARENA, for evaluating LLM-based agents on realistic, execution-time network/system tasks. Instead of relying on small, static, possibly contaminated benchmarks, it dynamically generates tasks over a unified state–action interface, runs them in emulators (Mininet/K8s/DC simulator), and scores agents on correctness, safety, and latency. Experiments across several network-style applications show current LLM agents perform much worse in these realistic, dynamic settings than static benchmarks suggest.

**Strengths:**

- Clear unified state–action abstraction that works across three concrete network apps (DC capacity planning, Mininet routing, K8s policy troubleshooting), not just a toy demo.
- Dynamic, on-demand query generation with stochastic sampling and emulator-backed ground truth, explicitly to cut contamination and widen coverage
- Execution-time evaluation on correctness, safety, and latency inside real emulators (Mininet, K8s, DC simulator), which exposes failure modes that static, correctness-only benchmarks miss

**Weaknesses:**

- RL/SFT “use cases” are proof-of-concept and on small models (Qwen2.5-0.5B, limited SFT splits), so the “can be used for rl training” claim is ahead of the evidence.
- All results are still in three networking-style environments; claims of generality beyond these domains are argued but not empirically shown.
- The dynamic generation relies on hand-designed templates and app-specific state equivalence/safety checks; portability to other operators’ emulators may be non-trivial.

**Questions:**

1. In 5.1 you show a GRPO run with Qwen2.5-0.5B in Mininet and note it “does not fully solve routing issues.” Can you clarify whether NETARENA currently supports stable, long-horizon RL runs (multiple episodes, curriculum, failure replay), or whether this is mainly a demonstration of feasibility? If it’s the latter, please make the scope explicit and report at least learning curves / success-per-episode to show the environment is not too sparse.

2. You claim the unified state–action abstraction “generalizes across applications,” but all experiments are DC capacity planning, Mininet routing, and K8s policy. Can you point to a non-network/system domain where you tried to plug in the same pipeline?

---

> ### Author Response · Authors · 2025-11-21
>
> Dear Reviewer,
>
> Thank you for your constructive feedback. Below, we address your comments point by point.
>
> 1. **More details of fine-tuning with RL**
> NetArena can be used for RL training because it provides (1) unlimited test sample generation, (2) an interactive environment, and (3) reward feedback based on accuracy, safety, and other metrics. **With the additional rebuttal pages, we add Figure 6 for more detailed analysis**.
>
> - **Figure 6(a):** The RL training curve is initially noisy for the first 36 episodes as the agent explores. After that, the cumulative reward steadily increases, indicating successful learning from environment feedback.
> - **Figure 6(b):** A qualitative before/after comparison.
>   - Before RL fine-tuning, Qwen-0.5B (CoT) outputs random and invalid commands for the Mininet emulator (e.g., LLM agent output:  `try to ping`).
>   - After fine-tuning, LLM agent outputs valid diagnostic and modification commands (e.g., `ping -c 3 192.168.1.2`).
>
> While overall correctness remains limited, these preliminary results imply that the environment provides learnable feedback. We have clarified this point in the updated paper draft.
>
> ---
>
> 2. **Generalization of “State–Action” abstraction**
> NetArena models the environment as states and structural edits in the environment as actions. This abstraction could extend to other domains with structured topology or dependency graphs. We outline two examples of how this might be done in the table below for discussion.
>
> | State (Nodes)         | State (Edges)                                                       | Action                                               |
> |-----------------------|----------------------------------------------------------------------|------------------------------------------------------|
> | Capacity Planning (in NetArena) | Datacenter devices                                                  | Physical/logical links (connection status, bandwidth) | Add/remove devices, adjust capacity, aggregate usage |
> | Routing (in NetArena) | Routers                                                              | Routing/forwarding status (e.g., ping mesh, link reachability) | Diagnose, drop/disable interface                  |
> | K8s (in NetArena)     | Microservice deployment                                              | Service links (e.g., connectivity)                    | Add/delete ingress rules, change protocol, patch deployments |
> | **IoT Management (new)**  | IoT sensors, gateways                                                | Connectivity links (e.g., signal strength), data-flow paths | enable/disable sensors, update devices           |
> | **Robotics (new)**       | Robot joints, environment objects                                    | Joint articulation, object contacts                   | Change joint-angle, move object to target          |
>
> ---
>
> 3. **Extending NetArena to other operators’ emulators**
> To support a new emulator, domain experts only need to define the State representation, the Action space, and task templates once. Appendix Table 3 (page 16) provides step-by-step instructions for extending NetArena. For the three evaluated applications, the required effort includes:
>
> (1) state extraction (less than 150 lines of code),
> (2) action definition (less than 500 lines of code),
> (3) correctness/safety checking (less than 80 lines of code).
>
> For applications lacking established emulators, recent work [1] shows that LLMs combined with formal methods can automatically synthesize emulators. Integrating such techniques is a natural next step to expand NetArena to a broader set of real-world environments.
>
> We are actively expanding NetArena to more models, agents and applications with evaluators. Please see the newly added results with 11 models and agents on the anonymized leaderboard: https://alphabetsoup628.github.io/netarena_leaderboard/
>
> **[1]** Archit Bhatnagar, Yiming Qiu, Sarah McClure, Sylvia Ratnasamy, and Ang Chen. 2025. *A Case for Learned Cloud Emulators*. In Proceedings of the 24th ACM Workshop on Hot Topics in Networks (HotNets '25). Association for Computing Machinery, New York, NY, USA, 69–76. https://doi.org/10.1145/3772356.3772404

---

> ### Author Response · Authors · 2025-11-26
>
> Dear Reviewer kJeZ,
>
> Thank you again for the detailed review! Just for your information, we added more details for the RL-based fine-tuning use case to the paper draft, including the new Figure 6. All the changes are marked in blue for your convenience.
>
> Please feel free to let us know if you have any more questions or concerns! We would love to address them further.
>
> Best wishes to your work!
>
> -NetArena Authors

---

### Official Review · Reviewer_7kRM · 2025-11-05

**Soundness:** 3
**Presentation:** 3
**Contribution:** 4
**Rating:** 6
**Confidence:** 3

**Summary:**

The paper presents NETARENA, a dynamic benchmarking framework for evaluating LLMs in realistic network and system environments. It addresses critical limitations of existing static benchmarks, including data contamination risks, high statistical variance from limited dataset sizes, and inadequate representation of production environment complexity.. The framework defines a unified state–action abstraction that enables automatic query and ground truth generation across applications such as datacenter capacity planning, routing misconfiguration, and microservice policy troubleshooting. By integrating high-fidelity network emulators like Mininet and Kubernetes, NETARENA provides runtime feedback on correctness, safety, and latency. Experiments with models such as GPT-4o and Qwen-72B demonstrate low average correctness (13–38%), underscoring the complexity of real-world network tasks. NETARENA also supports supervised fine-tuning and reinforcement learning, enabling scalable, dynamic, and contamination-resistant evaluation of LLM agents in safety-critical network operations.

**Strengths:**

1. This paper effectively solves data contamination risk through dynamic generation, eliminates statistical unreliability of small datasets , and captures real-world complexity missing in existing benchmarks
1. It integrates with production-grade emulators (Mininet, Kubernetes), and provides execution-grounded assessment beyond simple correctness, including safety and latency metrics.
1. It supports 9,250+ queries with unlimited generation, while maintaining diversity across complexity levels and task types that static benchmarks cannot achieve.
1. The framework provides an environment that supports RL training and evaluation of LLMs in realistic network applications.

**Weaknesses:**

1. Limited agent diversity: The evaluation only includes baseline prompting strategies (CoT, Few-shot, ReAct), which may not fully represent the capabilities of advanced LLM-based agents in network reasoning tasks.
1. The integration with high-fidelity emulators may introduce significant setup challenges, potentially reducing the reproducibility and accessibility of the framework.
1. While correctness, safety, and latency are meaningful metrics, the evaluation could be enriched with additional dimensions.
1. Although RL post-training is mentioned, the paper does not include experimental results or analysis for RL-based fine-tuning.

Minors:
1. QWen -> Qwen

**Questions:**

1. What is the complexity of running the emulators and evaluation process at scale?
1. How is the ground truth constructed for the SFT dataset described in Section 4.3?

---

> ### Author Response · Authors · 2025-11-21
>
> Dear Reviewer,
>
> Thank you for your constructive feedback. Below, we address your comments point by point.
>
> 1. **Is integration with emulators hard to set up?**
> No. All emulators in NetArena are fully containerized. The entire system runs end-to-end with a single Docker command, ensuring automatic installation and fully reproducible configurations. The anonymized README provides exact setup instructions:  https://anonymous.4open.science/r/netarena_iclr2026-BE94/README.md
>
> ---
>
> 2. **How is ground truth constructed for the SFT dataset (Section 4.3)?**
> For constructive tasks such as capacity planning, each query defines a transformation from the initial state `S_0` to a target state `S_n`. NetArena automatically synthesizes the complete action sequence `a1, a2, ..., an` that reaches `S_n`. We use these dynamically generated action traces as ground-truth supervision signals for SFT.
> Below is a concrete query and ground truth example used in the SFT datasets.
>
> **Query:**  "Add a new_packet_switch_27 to ju1.s2.dom. Count the packet_switch in ju1.s2.dom in the updated graph and return the total capacity."
>
> **Ground truth:**
> ```python
> def solution(graph_data=datacenter_topology):
>     # a1: add a new switch
>     graph_data = action_add_node_to_graph(graph_data, new_node, parent_node_name)
>
>     # a2: count the switch
>     count = action_counting(graph_data, node1, node2)
>
>     # a3: calculate the total capacity
>     capacity = action_sum_capacity(graph_data, node1, attr='capacity')
>
>     return count, capacity
> ```
>
> Since each `action_type` has a predefined structure, only the dynamic parameters and attributes need to be filled in. This automatically generated ground truth guarantees correctness and can generate arbitrarily large SFT datasets.
>
> ---
>
> 3. **Lack of RL-based fine-tuning results.**
> Our RL-based fine-tuning experiments serve as a proof-of-concept showing that NetArena can support interactive, environment-driven feedback for policy improvement. **With the additional rebuttal pages, we add Figure 6 for more detailed analysis**.
>
> - **Figure 6(a):** The RL training curve is initially noisy for the first 36 episodes as the agent explores. After that, the cumulative reward steadily increases, indicating successful learning from environment feedback.
> - **Figure 6(b):** A qualitative before/after comparison.
>   - Before RL fine-tuning, Qwen-0.5B (CoT) outputs random and invalid commands for the Mininet emulator (e.g., LLM agent output:  `try to ping`).
>   - After fine-tuning, LLM agent outputs valid diagnostic and modification commands (e.g., `ping -c 3 192.168.1.2`).
>
> While overall correctness remains limited, these results demonstrate that the environment supplies non-sparse, learnable feedback.
>
> ---
>
> 4. **Limited agent diversity in current experiments.**
> We are actively expanding model and agent coverage. We have added a state-of-the-art network domain-specific agent, **MeshAgent [1]**, which uses program synthesis to improve network-management task performance, as well as two new LLMs (Gemini-2.5-Pro, Claude-4-Sonnet). The table below provides updated results on the Capacity Planning task.
> Please see the newly added results with 11 models and agents on the anonymized leaderboard:
> https://alphabetsoup628.github.io/netarena_leaderboard/
>
> | Model                        | Avg. Correctness (95% CI) | Avg. Safety (95% CI) | Avg. Latency |
> |-----------------------------|----------------------------|-----------------------|---------------|
> | Gemini-2.5-Pro (CoT)        | 38% [0.01, 0.03]           | 52% [0.02, 0.05]      | 14s           |
> | Claude-4-Sonnet (CoT)       | 46% [0.01, 0.02]           | 58% [0.02, 0.06]      | 10s           |
> | Gemini-2.5-Pro (MeshAgent)  | 59% [0.01, 0.03]           | 66% [0.01, 0.04]      | 20s           |
> | Claude-4-Sonnet (MeshAgent) | 74% [0.01, 0.02]           | 70% [0.03, 0.06]      | 24s           |
> | GPT-4o (MeshAgent)          | 68% [0.01, 0.06]           | 75% [0.05, 0.07]      | 22s           |
> | QWen-72B (MeshAgent)        | 55% [0.01, 0.05]           | 68% [0.08, 0.15]      | 29s           |
>
> ---
>
> **[1]** Zhou, Y., Hsieh, K., Mani, S.K., Kandula, S. and Liu, Z., 2025, December. *MeshAgent: Enabling Reliable Network Management with Large Language Models*. ACM SIGMETRICS.

---

> ### Author Response · Authors · 2025-11-26
>
> Dear Reviewer 7kRM,
>
> Thank you again for the detailed review! Just for your information, we added more details for the RL-based fine-tuning use case to the paper draft, including the new Figure 6. All the changes are marked in blue for your convenience.
>
> Please feel free to let us know if you have any more questions or concerns! We would love to address them further.
>
> Best wishes to your work!
>
> -NetArena Authors

---

### Author Response · Authors · 2025-12-02

Dear AC,

Thank you! We appreciate your extra time during the reviewing process.


Since we had limited interaction with the reviewers, we summarize the key concerns and our responses below. Please see more details in the rebuttal.


| **Reviewer Comments** | **Our Response Summary** |
|----------------------|---------------------------|
| Details of RL-based fine-tuning with NetArena | NetArena naturally supports RL because it provides (1) unlimited test-sample generation, (2) an interactive environment, and (3) reward signals grounded in accuracy, safety, and other metrics. We added Figure 6 and additional analysis in the paper revision draft for clarity. |
| Generalizability of NetArena’s dynamic benchmark generation | Our “State–Action” abstraction generalizes broadly to applications with underlying graph topologies (e.g., Robotics, IoT). Operators simply define the State and Action spaces, and with an emulator, NetArena can generate unlimited new queries on demand. |
| Ease of running NetArena | Yes. The framework is fully open-sourced (https://anonymous.4open.science/r/netarena_iclr2026-BE94/README.md), supports Docker deployment. We are actively expanding to more real-world applications and domain-specific agent types (https://alphabetsoup628.github.io/netarena_leaderboard/). |
| | |

In short, as LLM benchmarks continue to grow, we want to highlight the overlooked risks of static benchmark contamination and the need to evaluate agents under realistic deployment conditions.  We believe NetArena offers a timely milestone for the community, both as a practical evaluation tool for network and system agents and as a general methodology for dynamic benchmark generation.


Wish you all the best!


Warm regards,
NetArena Authors

---

### Meta-Review · Area_Chair_GdM5 · 2026-01-06

**Summary:**

This paper introduces NetArena which is for evaluating LLM-based agents in realistic network. This allows for on-demand task generation and execution-grounded scoring, which solves limitations of static benchmarks. Reviewers generally think the submission is timely and well presented. Existing models are shown to be less effective under these settings. The overall ratings are 6,6,6. AC recommends accept.

**Reviewer Concerns:**

The main concerns are limited agent diversity and the scope of RL/SFT. The authors' rebuttal looks convincing. The RL/SFT results are positioned as proof of concept, which looks appropriate.

**Reviewer Scores:**

Reviewers would've raised their scores if they had had the opportunity to participate in full discussion.

---

### Decision · Program_Chairs · 2026-01-26

Accept (Poster)